# Novel Variance-Component TWAS method for studying complex human diseases with applications to Alzheimer's dementia

Shizhen Tang[1,2], Aron S. Buchman[3], Philip L. De Jager[4], David A. Bennett[3], Michael P. Epstein[1], Jingjing Yang[1] *

1 Center for Computational and Quantitative Genetics, Department of Human Genetics, Emory University School of Medicine, Atlanta, Georgia, United States of America, 2 Department of Biostatistics and Bioinformatics, Emory University School of Public Health, Atlanta, Georgia, United States of America, 3 Rush Alzheimer's Disease Center, Rush University Medical Center, Chicago, Illinois, United States of America, 4 Center for Translational and Computational Neuroimmunology, Department of Neurology and Taub Institute for Research on Alzheimer's Disease and the Aging Brain, Columbia University Irving Medical Center, New York, New York, United States of America

* jingjing.yang@emory.edu

**Data Availability Statement:** All data used in this paper are available either from public website or requested from the following websites. RADC

## Abstract

Transcriptome-wide association studies (TWAS) have been widely used to integrate transcriptomic and genetic data to study complex human diseases. Within a test dataset lacking transcriptomic data, traditional two-stage TWAS methods first impute gene expression by creating a weighted sum that aggregates SNPs with their corresponding cis-eQTL effects on reference transcriptome. Traditional TWAS methods then employ a linear regression model to assess the association between imputed gene expression and test phenotype, thereby assuming the effect of a cis-eQTL SNP on test phenotype is a linear function of the eQTL's estimated effect on reference transcriptome. To increase TWAS robustness to this assumption, we propose a novel Variance-Component TWAS procedure (VC-TWAS) that assumes the effects of cis-eQTL SNPs on phenotype are random (with variance proportional to corresponding reference cis-eQTL effects) rather than fixed. VC-TWAS is applicable to both continuous and dichotomous phenotypes, as well as individual-level and summary-level GWAS data. Using simulated data, we show VC-TWAS is more powerful than traditional TWAS methods based on a two-stage Burden test, especially when eQTL genetic effects on test phenotype are no longer a linear function of their eQTL genetic effects on reference transcriptome. We further applied VC-TWAS to both individual-level (N = ~3.4K) and summary-level (N = ~54K) GWAS data to study Alzheimer's dementia (AD). With the individual-level data, we detected 13 significant risk genes including 6 known GWAS risk genes such as *TOMM40* that were missed by traditional TWAS methods. With the summary-level data, we detected 57 significant risk genes considering only cis-SNPs and 71 significant genes considering both cis- and trans- SNPs, which also validated our findings with the individual-level GWAS data. Our VC-TWAS method is implemented in the TIGAR tool for public use.

Research Resource Sharing Hub, http://www.radc.rush.edu/; ROS/MAP data, https://www.synapse.org/#!Synapse:syn3219045; MayoClinicLOAD data, https://www.synapse.org/#!Synapse:syn2910256; IGAP summary data, https://web.pasteur-lille.fr/en/recherche/u744/igap/igap_download.php.

**Funding:** ST and JY are supported by National Institutes of Health (NIH/NIGMS) grant award R35GM138313. MPE was supported by NIH/NIGMS grant award R01GM117946 and NIH/NIA grant award RF1AG071170. Data collection was supported through funding by NIH/NIA grants P30AG10161, R01AG15819, R01AG17917, R01AG30146, R01AG36836, R01AG56352, U01AG32984, U01AG46152, U01AG61356, the Illinois Department of Public Health, and the Translational Genomics Research Institute. These grants support the generation of the ROS/MAP data, which is led by ASB, PLDJ and DAB. The funders had no role in study design, data collection and analysis, decision to publish, or preparation of the manuscript.

**Competing interests:** The authors have declared that no competing interests exist.

## Author summary

Traditional Transcriptome-wide association studies (TWAS) tools make strong assumptions about the relationships among genetic variants, transcriptome, and phenotype that may be violated in practice, thereby substantially reducing the power. Here, we propose a Variance-Component TWAS method (VC-TWAS) that relaxes these assumptions and can be implemented with both individual-level and summary-level GWAS data, which is suitable for studying both continuous and dichotomous phenotypes. Our simulation studies showed that VC-TWAS achieved higher power compared to traditional TWAS methods based on a two-stage Burden test, when the underlying assumptions required by traditional TWAS tools were violated. We further applied VC-TWAS to both individual-level (N = ~3.4K) and summary-level (N = ~54K) GWAS data to study Alzheimer's dementia (AD). With individual-level data, we detected 13 significant risk genes including 6 known GWAS risk genes such as *TOMM40* that were missed by traditional TWAS methods. Interestingly, 5 of these genes were shown to possess significant pleiotropic effects on AD pathology phenotypes, revealing possible biological mechanisms. With summary-level data of a larger sample size, we detected 57 significant risk genes considering only cis-SNPs and 71 significant genes considering both cis- and trans- SNPs, which also validated our findings with the individual-level GWAS data. In conclusion, VC-TWAS provides an important analytic tool for identifying risk genes whose effects on phenotypes might be mediated through transcriptomes.

## Introduction

Genome-wide association studies (GWAS) have succeeded in identifying thousands of genetic loci associated with complex traits and diseases [1–3]. However, for the most part the molecular mechanisms linking these genes with these complex traits and diseases remain unexplained [4]. Studies have shown that gene expression plays a key role in the phenotypic manifestation of human diseases [5]. Many common genetic variants associated with the phenotypes manifested by human diseases are highly likely to be expression quantitative trait loci (eQTL) [6,7]. Therefore, integrating gene expression data together with genetic data from GWAS is expected to help identify novel risk genes as well as elucidate the mechanisms underlying the associations of genetic loci with complex traits and disease phenotypes.

 The transcriptome-wide association study (TWAS) is an innovative strategy to leverage the enhanced power of integrating gene expression of the transcriptome together with genetic data from GWAS [8] for gene-based association studies of complex traits. Using a reference transcriptomic panel like Genotype-Tissues Expression (GTEx) [9], traditional TWAS methods first train a regression model that treat gene expression as the outcome and SNP genotype data (generally cis-SNPs nearby the test gene) as predictors, which can be viewed as a gene expression imputation model. traditional two-stage TWAS tools such as PrediXcan [10], FUSION [11], and TIGAR [8] employ different regression methods to fit such gene expression imputation models, where corresponding cis-eQTL effect sizes are derived. For example, PrediXcan method [10] implements the Elastic-Net penalized regression method [12] while TIGAR [8] implements the nonparametric Bayesian Dirichlet process regression (DPR) [13] method. Regardless of technique, these existing two-stage TWAS methods produce a set of estimates of cis-eQTL effects on reference transcriptome. Within a GWAS lacking transcriptomic data, existing two-stage TWAS tools then imputes genetically-regulated gene expression

(GReX) by weighted summing cis-eQTL SNP data from the GWAS with corresponding estimates of cis-eQTL effects on reference transcriptome. Once created, existing two-stage TWAS tools test for association between the imputed GReX and traits of interest using GWAS data, where the test is based on a single variant linear regression model. TWAS methods have been deployed widely since their inception and have had substantial success in improving our understanding the genetic regulation of various complex traits [14].

Since existing two-stage TWAS tools derive GReX as the summation of eQTL SNP genotypes weighted by their corresponding effects on reference transcriptome, they essentially conduct weighted burden tests [8] to evaluate the association between GReX and outcome (referred to as Burden-TWAS in this paper). Burden-TWAS methods assume the effects of eQTL SNPs on phenotype is a linear function of their corresponding estimated effects on reference transcriptome (see Methods), which may not be true in practice. In particular, the effect of eQTL SNP on phenotype may be a non-linear, rather than a linear, function of the eQTL effect on reference transcriptome. Moreover, the assumed eQTL effects on transcriptome in the test dataset could be mis-specified due to ancestral differences between the test and reference datasets. For example, recent studies show that gene-expression prediction models trained in one population perform poorly when applied to a different population [15]. The non-linearity assumption about the effects of eQTL SNPs on the phenotype of interest does not affect the validity of TWAS but it can reduce the power of these traditional two-stage Burden-TWAS methods.

Here, we derive a Variance-Component TWAS (VC-TWAS) method that relaxes the linearity assumption of Burden-TWAS. This modification makes this method more robust to misspecification of eQTL effect size estimates derived from the reference transcriptome, thus improving its power compared to the Burden-TWAS methods. VC-TWAS is a TWAS analogue of the Sequence Kernel Association Test (SKAT) [16–19] used for gene-based association studies. Unlike Burden-TWAS methods, our VC-TWAS aggregates genetic information across test SNPs using a kernel similarity function that allows upweighting or downweighing of specific variants in the similarity score based on eQTL effect size magnitudes (which can be derived using DPR methods in TIGAR or Elastic-Net penalized regression methods implemented in PrediXcan). The test statistic can be thought of as a Variance-Component score statistic based on a mixed model where each test variant has a random effect whose variance is a linear function of the squared values of corresponding eQTL effect size.

By modeling variants with random effects, the technique relaxes the main assumption of Burden-TWAS and is robust to misspecification of eQTL effect size estimates derived from the reference transcriptome (in terms of the direction and magnitude). The variance component score test employed by VC-TWAS also enjoys efficient computation and flexibility for studying both quantitative and dichotomous phenotypes, which enables VC-TWAS to consider both cis- and trans- eQTL SNPs. VC-TWAS is further applicable to both individual-level and summary-level GWAS data.

We note that VC-TWAS is not the first TWAS method developed that relaxes the linearity assumption of Burden-TWAS. A recently derived collaborative mixed model (CoMM) [20,21] likewise accounts for the uncertainty of eQTL effect size estimates from reference transcriptome data by jointly modeling reference and test data within a linear mixed-model framework. The PMR-Egger [22] and moPMR-Egger [23] also take the approach of jointly modeling reference and test data to address the uncertainty of eQTL effect size estimates from the reference data. However, these methods that are based on the maximum likelihood inference framework and implement likelihood ratio tests, are derived only for quantitative phenotypes and could suffer computation burden for testing thousands of cis-SNPs per gene, which often happens in practice particularly when considering imputed SNP data or whole-genome sequencing data for TWAS. In this study, we show that the likelihood ratio test approach used by CoMM and

PMR-Egger is computationally more expensive to run than VC-TWAS in practice when there are thousands of test SNPs such as using imputed or Whole Genome Sequencing (WGS) genotype data.

This manuscript is as organized as follows. First, we provide an overview of traditional two-stage Burden-TWAS as well as existing techniques for estimating cis-eQTL SNP effects from reference transcriptome data. We then derive VC-TWAS approach for using both individual-level and summary-level GWAS data and compare its performance with Burden-TWAS using simulated data generated under different eQTL model assumptions. We then apply VC-TWAS to individual-level GWAS data from Religious Order Study and Memory Aging Project (ROS/ MAP) [24–27] and Mayo Clinic Late-Onsite Alzheimer's disease (LOAD) [28,29] cohorts, as well as the summary-level GWAS data from the International Genomics of Alzheimer's Project (IGAP) [3], for studying Alzheimer's dementia (AD). By considering only cis-eQTL effect sizes, VC-TWAS identified both novel and known risk genes for AD within 2MB of the well-known major risk gene *APOE* of AD, including the known risk gene *TOMM40* and *APOE*. Considering both cis- and trans- eQTL effect sizes estimated by the Bayesian Genome-wide TWAS (BGW-TWAS) method [30], VC-TWAS detected 71 risk genes for AD which complemented existing TWAS results using only cis-eQTL SNP data. After describing the results, we provide a brief discussion summarizing our findings and describing implementation of VC-TWAS into our previously developed software tool TIGAR [8] for public use.

## Methods

### Ethics statement

Real ROS/MAP and Mayo Clinic GWAS data analyzed in this study were generated under the improvement by the Institutional Review Board (IRB) of Rush University Medical Center, Chicago, IL and Mayo Clinic, respectively. All samples analyzed in this study were de-identified and all analyses were approved by the IRB of Emory University School of Medicine.

### Traditional Two-Stage TWAS procedure

Two-stage TWAS first fits gene expression imputation models by taking genotype data as predictors and assuming the following additive genetic model for expression quantitative traits,

$$E_g = Gw + \varepsilon, \varepsilon \sim N(0, \sigma_\epsilon^2 I). \tag{1}$$

Here, $G$ is the genotype matrix for all considered SNP genotypes (encoded as the number of minor alleles or genotype dosages of SNPs within 1MB of the target gene region), $w$ is the eQTL effect size vector, and $E_g$ is the profiled gene expression levels for the target gene $g$. Given the eQTL effect size estimates $\hat{w}$ from reference data, $GReX$ will be imputed by the following equation

$$\widehat{GReX} = G_{new}\hat{w}, \tag{2}$$

where $G_{new}$ is the genotype matrix for the test cohort.

The general test framework of Burden-TWAS [8,10,11] that test for association between $\widehat{GReX}$ and the phenotype of interest can be written as:

$$E[g(Y|G_{new})] = \beta\widehat{GReX} + \alpha'Z = \beta(G_{new}\hat{w}) + \alpha'Z, \tag{3}$$

where $Y$ denotes the phenotype of interest, $g(.)$ denotes a link function, $\widehat{GReX}$ is imputed gene expression levels, and $\alpha'$ denotes the coefficient vector for other non-genomic covariates $Z$.

Basically, Burden-TWAS tests the null hypothesis of $H_0$: $\beta = 0$, where eQTL effect size estimates ($\hat{w}$) are taken as variant weights and SNP effect sizes on phenotype ($\beta\hat{w}$) are assumed to be a linear function of $\hat{w}$ [8,10,11]. As noted above this linear relationship between eQTL effect sizes on reference transcriptome and the SNP effect sizes on test phenotype may not be true when analyzing human data and limits the power of the existing two-stage Burden-TWAS methods.

## Estimation of eQTL effect sizes

Different methods can be used to estimate eQTL effect sizes $w$ from Eq (1). In this study, we applied PrediXcan (Elastic-Net penalized regression) and TIGAR (nonparametric Bayesian DPR methods) [8,10] to estimate $w$ that only consider cis-eQTL. PrediXcan TWAS method [10] employs Elastic-Net penalized regression method [12] to estimate cis-eQTL effect sizes $w$ from Eq (1) (S1A Text). TIGAR[8] provides a more flexible approach to nonparametrically estimate cis-eQTL effect sizes by a Bayesian DPR method [13] (S1B Text). Additionally, we also considered modeling gene expression using both cis- and trans- eQTL effect sizes estimated by the recently proposed Bayesian genome-wide TWAS (BGW-TWAS) method [30].

## VC-TWAS with individual-level GWAS data

Here, we propose a powerful VC-TWAS method that is analogous to the previously proposed SKAT method for SNP-set based association studies [16]. Similar to SKAT, the general test framework of VC-TWAS can be written as

$$Y_i = \boldsymbol{\beta}' \boldsymbol{G_i} + \boldsymbol{\alpha}' \boldsymbol{Z_i} + \boldsymbol{\varepsilon}_i, \ \beta'_j \sim N(0, w_j^2 \tau), \varepsilon_i \sim N(0, \sigma_\epsilon^2),$$

for continuous quantitative traits, and

$$logit(Prob(Y_i = 1)) = \boldsymbol{\beta}' \boldsymbol{G_i} + \boldsymbol{\alpha}' \boldsymbol{Z_i} + \boldsymbol{\varepsilon}_i, \beta'_j \sim N(0, w_j^2 \tau), \ \varepsilon_i \sim N(0, \sigma_\epsilon^2),$$

for dichotomous traits of sample $i$. Here, $\boldsymbol{\beta}$ is the genetic effect size vector, $\boldsymbol{G}$ is the genotype matrix for all test SNPs with respect to the test gene, $\boldsymbol{Z}$ is the non-genomic covariate matrix, and $\boldsymbol{\varepsilon}$ is the error term. VC-TWAS will test $H_0$: $\tau = 0$, which is equivalent to testing $H_0$: $\boldsymbol{\beta} = 0$. The Variance-Component score statistic used by VC-TWAS is given by

$$Q = (\boldsymbol{Y} - \widehat{\boldsymbol{\mu}})' \boldsymbol{K} (\boldsymbol{Y} - \widehat{\boldsymbol{\mu}}), \boldsymbol{K} = \boldsymbol{GWG}', \tag{4}$$

where $\widehat{\boldsymbol{\mu}}$ is the estimated phenotype mean under $\boldsymbol{H_0}$ and $\boldsymbol{W} = diag(w_j^2, \ldots)$ with weight $w_j$ for the $j$th variant.

In contrast to SKAT methods, VC-TWAS takes eQTL effect size estimates from Eq (1) as variant weights ($w_j$). That is, the variances ($\tau w_j^2$) of SNP effect sizes on phenotype are assumed to be a linear function of eQTL effect size estimates, which is robust to both direction and magnitude of eQTL effect size estimates. Since the Variance-Component score statistic $Q$ (Eq (4)) follows a mixture of chi-square distributions under the null hypothesis [31,32], p-value can be conveniently obtained from several approximation and exact methods like the Davies exact method [33].

We note that our VC-TWAS method is computationally more expensive than standard Burden-TWAS given the need to perform eigen-decomposition of the kernel matrix $\boldsymbol{K}$ in Eq 4 to obtain an analytic p-value (S1C Text). Such eigen-decomposition has computational complexity $O(m^3)$ for considering $m$ SNPs with non-zero eQTL effect sizes. As DPR method produces non-zero cis-eQTL effect size estimates for almost all test SNPs (with most cis-eQTL effect size estimates being close to zero [8]), we explored an alternate VC-TWAS that

considered a reduced set of SNPs by filtering out those with cis-eQTL effect size estimates smaller than the median cis-eQTL effect size estimate per gene. As we show using simulated data, we can reduce up to 80% computation time while having negligible impact on performance relative to using all SNPs with non-zero cis-eQTL effect size estimates.

Compared to Burden-TWAS, VC-TWAS is based on a random effect model and test for if the variation of random effects differs from 0. That is, the variance component score statistic (Eq (4)) used by VC-TWAS does not directly model the directions of cis-eQTL effect sizes ($w_j$) and is robust for mis-specifications of cis-eQTL effect sizes.

## VC-TWAS with summary-level GWAS data

Since summary-level GWAS data generally provide SNP effect sizes on phenotype and corresponding standard errors based on single variant tests (often meta-analysis), it is reasonable to assume that the test phenotypes were adjusted for the confounding covariates. That is, in the $Q$ statistic used by VC-TWAS as in Eq (4), we can assume the phenotype mean $\widehat{\boldsymbol{\mu}}$ under $\boldsymbol{H_0}$ is 0. This leads to a simplified formula [34,35]

$$Q = \sum_{j=1}^{m} w_j^2 s_j^2, \tag{5}$$

where $s_j = \boldsymbol{G}_j' \boldsymbol{Y}/\widehat{\sigma_Y}^2$ is the single variant score statistic of the $j^{th}$ variant, $\widehat{\sigma_Y}^2$ is the estimated phenotype variance.

As derived in the previous studies [36], we can estimate phenotype variance and then the score statistics by using only GWAS summary statistics including the single variant effect size estimate $\widehat{\beta}_j$ and corresponding standard error $\widehat{\sigma}_j$ for $j^{th}$ SNP, sample size $n$, and a reference LD covariance matrix $\Sigma$ of all test SNPs (S1C Text). That is, the numerator and denominator of the score statistic can be estimated by,

$$\boldsymbol{G}_j' \boldsymbol{Y} = (n-1)\widehat{\beta}_j \Sigma_{j,j}; \widehat{\sigma_Y}^2 = median(\Sigma_{j,j}\widehat{\sigma}_j^2(n-1) + \Sigma_{j,j}\widehat{\beta}_j^2; j = 1, \ldots, m). \tag{6}$$

## ROS/MAP data

In this study we used clinical and postmortem data from older adults participating in two ongoing community-based cohorts studies the Religious Orders Study (ROS) and Rush Memory and Aging Project (MAP) [24–27] which document risk factors and chronic conditions of aging including dementia. Participants are senior adults without known dementia, who agree to annual clinical evaluation and brain autopsy at the time of death. All participants signed an informed consent and Anatomic Gift Act, and the studies are approved by an Institutional Review Board of Rush University Medical Center, Chicago, IL. All participants in this study also signed a repository consent to allow their data to be re-purposed.

Currently, we have microarray genotype data generated for 2,093 European-decent subjects from ROS/MAP [24–27], which are further imputed to the 1000 Genome Project Phase 3 [37] in our analysis [38]. Post-mortem brain samples (gray matter of the dorsolateral prefrontal cortex) from ~30% these ROS/MAP participants with assayed genotype data are also profiled for transcriptomic data by next-generation RNA-sequencing [39], which are used as reference data to train GReX prediction models in our application studies.

Using ROS/MAP data, we conducted TWAS for clinical diagnosis of late on-site Alzheimer's dementia (LOAD) as well as three pathology indices of AD including quantified $\beta$-amyloid load and PHFtau tangle density as well as a summary measure of AD pathology burden (S1D Text) [24,25,27].

## Mayo Clinic LOAD GWAS data

Mayo Clinic LOAD GWAS data contain samples from two clinical AD Case-Control series: Mayo Clinic Jacksonville (MCJ: 353 AD cases and 331 Controls), Mayo Clinic Rochester (MCR: 245 AD cases and 701 Controls) and a neuropathological series of autopsy-confirmed subjects from the Mayo Clinic Brain Bank (MCBB: 246 AD cases and 223 non-AD Controls) [28,29]. In total, we have 844 cases with LOAD and 1,255 controls without a dementia diagnosis. Mayo Clinic LOAD GWAS data have microarray genotype data profiled for 2,099 European-decent samples that are further imputed to the 1000 Genome Project Phase 3 [37] in our analysis [38]. This cohort only profiles the phenotype of clinical diagnosis of AD.

## IGAP GWAS summary statistics of AD

We used the GWAS summary statistics of AD generated from the stage 1 of the International Genomics of Alzheimer's Project (IGAP) with individuals of all European ancestry [3]. This summary-level GWAS data was generated by meta-analysis (N = ~54K) of four previously-published GWAS datasets consisting of 17,008 Alzheimer's disease cases and 37,154 controls (The European Alzheimer's disease Initiative–EADI the Alzheimer Disease Genetics Consortium–ADGC The Cohorts for Heart and Aging Research in Genomic Epidemiology consortium–CHARGE The Genetic and Environmental Risk in AD consortium–GERAD).

## Simulation study design

The purpose of this simulation study is to compare the performance of Burden-TWAS and VC-TWAS with variant weights estimated by PrediXcan and DPR methods, as well as validate the VC-TWAS approach of using only summary-level GWAS data. We used the real genotype data from ROS/MAP [40] participants to simulate quantitative gene expression and phenotype traits, where the genotype data were of 2,799 cis-SNPs (with $MAF>5\%$ and Hardy Weinberg p-value $>10^{-5}$) of the arbitrarily chosen gene $ABCA7$.

Specifically, quantitative gene expression traits are generated by the following equation

$$E_g = Gw + \varepsilon_E, \tag{7}$$

where $G$ denotes the genotype matrix of randomly selected true causal eQTL based on a target proportion of causal eQTL ($p_{causal}$) within the test gene, $w$ denotes cis-eQTL effect sizes generated from $N(0, \sigma_w^2 I)$ with variance $\sigma_w^2$ chosen to ensure a target gene expression heritability ($h_e^2$), and $\varepsilon_E$ is the error term generated from $N(0, (1 - h_e^2)I)$.

Phenotype data are generated based on two models to mimic two different genetic architectures of complex traits that may be encountered with human data:

**Model I:** The genetic effects on the phenotype of interest are completely driven by genetically regulated gene expression (GReX), where SNP effect sizes are of a linear function of their corresponding cis-eQTL effect sizes as assumed by Burden-TWAS methods [8,10,11]. Phenotype data are generated from the following equation

$$Y = \phi E_g + \varepsilon_Y = r(Gw + \varepsilon_E) + \varepsilon_Y, \varepsilon_Y \sim N(0, (1 - h_p^2)I), \tag{8}$$

where $E_g$ is the gene expression generated from Eq (7) and $r = \sqrt{h_p^2/Var(E_g)}$ is a scalar chosen to ensure a target phenotype variation proportion due to gene expression ($h_p^2$).

**Model II:** The magnitudes of SNP effect sizes on the phenotype are no longer a linear function of corresponding cis-eQTL effect sizes as in Model I, but rather are derived randomly from a distribution whose variance is a linear function of the squares of such effects. By doing

this, the SNP effect sizes on phenotype are a function of cis-eQTL effect sizes but the assumption of a linear relationship between the two (assumed by Burden-TWAS) is relaxed. Phenotype data are generated from the following equation

$$Y = G\beta + \varepsilon_Y, \varepsilon_Y \sim N(0, (1 - h_p^2)I), \tag{9}$$

where $G$ denotes the genotype matrix of randomly selected true causal SNPs that are also true causal cis-eQTL as in Eq (7), respective SNP effect sizes are generated from $\beta_i \sim N(0, rw_i^2)$ with corresponding cis-eQTL effect size $w_i$ as used in Eq (7) and $r$ chosen to ensure a phenotype heritability $h_p^2$.

We considered scenarios with various proportions of causal cis-eQTL/SNPs $p_{causal}$ = (0.001, 0.01, 0.1, 0.2) for the test gene, and various combinations of expression heritability ($h_e^2$) and phenotype variance proportion/heritability ($h_p^2$) that were chosen to ensure TWAS power falling within the range of (25%, 90%). The values of $p_{causal}$ and $h_p^2$ were taken as $(p_{causal}, h_p^2) = ((0.001, 0.2), (0.01, 0.3), (0.1, 0.4), (0.2, 0.5))$ for simulating phenotypes from Model I, while taken as $(p_{causal}, h_p^2) = ((0.001, 0.1), (0.01, 0.1), (0.1, 0.15), (0.2, 0.15))$ for simulating phenotypes from Model II.

We used 499 ROS/MAP samples as training data that were also used as training data in our application studies, and randomly selected 1,232 ROS/MAP samples as test data. To show the power performance with respect to test sample size, we considered different test sample sizes (400, 800, 1232) under the scenario with $p_{causal}$ = 0.2.

We estimated cis-eQTL effect sizes from training data by using PrediXcan and TIGAR DPR methods and then conducted Burden-TWAS and VC-TWAS with individual-level and summary-level test GWAS data. We note that Burden-TWAS using PrediXcan weights is equivalent to the PrediXcan method [10] while Burden-TWAS using DPR weights is equivalent to the TIGAR method [8]. We also compared Burden-TWAS and VC-TWAS with the CoMM method under simulation settings with $p_{causal}$ = 0.2 (we did not consider additional simulation settings because of the computational demands of CoMM).

For each scenario, we repeated simulations for 1,000 times and obtained the power as the proportion of simulations that had test p-value $<2.5\times10^{-6}$ (genome-wide significance threshold for gene-based test). Additionally, we simulated phenotype under the null hypothesis $Y\sim N(0,1)$ for $10^6$ times and evaluated type I errors of Burden-TWAS and VC-TWAS, using variant weights derived from PrediXcan and DPR methods.

For each scenario with VC-TWAS, we considered both the original form of the test as well as the alternate computationally-efficient form that considered only the filtered set of variants with cis-eQTL effect size estimates greater than the median effect size value in each simulation (about 50% SNPs).

## Results

### Simulation results

We compared the performance of VC-TWAS and Burden-TWAS using PrediXcan weights (cis-eQTL effect size estimates by Elastic-Net penalized regression) and DPR weights (cis-eQTL effect size estimates by DPR) under various scenarios. We also evaluated the performance of VC-TWAS and Burden-TWAS using filtered DPR weights as described in Methods. Under the scenario with $p_{causal}$ = 0.2, we considered different test sample sizes (400, 800, 1232), and compared VC-TWAS with CoMM using both individual-level and summary level test data with 1232 test samples.

First, we compared TWAS power for studying phenotypes simulated from Model I that assumed SNP effect sizes on phenotypes were of a linear function of their corresponding cis-

eQTL effect sizes. When $p_{causal}$ = (0.001, 0.01) with sparse true causal signals (S1A Fig), VC-TWAS had comparable power with Burden-TWAS and both TWAS methods using PrediXcan weights achieved higher power compared to using DPR weights. When $p_{causal}$>0.01, we observed that both TWAS methods using DPR weights achieved higher power compared to using PrediXcan weights. These results are consistent with the previous TIGAR paper [8]. This is because DPR method is preferred for modeling quantitative gene expression traits when a gene harbors a considerable proportion of true cis-eQTL with relatively smaller effect sizes, e.g., scenarios with $p_{causal}$>0.01 in our simulation studies. As expected, with $p_{causal}$>0.01 (S1A Fig, Fig 1A), Burden-TWAS methods outperformed VC-TWAS under Model I which meets the assumptions by Burden-TWAS methods. Across all considered scenarios, all TWAS methods using filtered DPR weights had similar performance as using complete DPR.

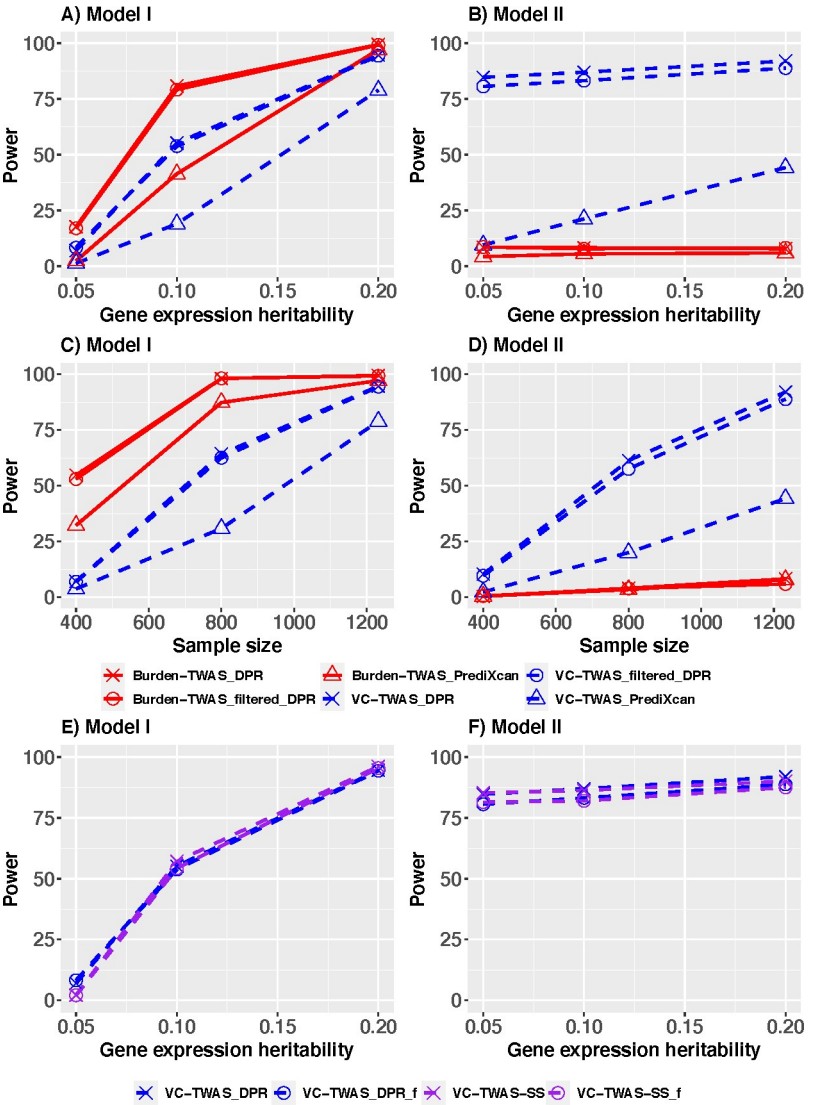

**Fig 1. Power comparison of Burden-TWAS and VC-TWAS methods under simulation scenarios with 20% true causal eQTL for gene expression (i.e., $p_{causal}$ = 0.2) in the test gene.** Phenotypes simulated from Model I (A, C, E) and Model II (B, D, F) were considered. Various gene expression heritability and various types of SNP weights were considered, including those derived from PrediXcan method, DPR method, and filtered DPR weights in panel (A, B). Various test sample sizes (400, 800, 1232) were considered in panel (B, D). The VC-TWAS approach of using only summary-level GWAS data were validated in panel (E, F).

Second, we compared TWAS power for studying phenotypes simulated from Model II that assumes variances of SNP effect sizes on phenotype were of a linear function of the squared values of their corresponding cis-eQTL effect sizes. As shown in Figs 1B and S1B, we found that the VC-TWAS obtained higher power than Burden-TWAS across all scenarios. Especially, with $p_{causal} \geq 0.01$, the power by VC-TWAS using DPR weights was double of that using PrediXcan weights on average (88.99% vs. 38.45%), while the power of Burden-TWAS using DPR weights was comparable with using PrediXcan weights except when $p_{causal} = 0.001$. When $p_{causal} = 0.001$ and $h_e^2 \in (0.1, 0.2)$ (S1B Fig), both TWAS approaches using PrediXcan weights performed better than using DPR weights. Again, across all considered scenarios, TWAS using filtered DPR weights had similar performance to that using complete DPR.

Third, we compared VC-TWAS and Burden-TWAS with different test sample sizes (400, 800, 1232) under the scenarios with ($p_{causal} = 0.2$, $h_e^2 = 0.2$). As shown in Figs 1C and 1D, the power of VC-TWAS and Burden-TWAS increased with respect to the test sample size in both simulation models. Interestingly, for model II, the magnitude of the power difference between VC-TWAS and Burden-TWAS became more pronounced with increasing sample size.

Fourth, to validate the VC-TWAS approach for using only summary-level GWAS data, we compared VC-TWAS with DPR weights by using individual-level and summary-level GWAS data, under the scenarios with $p_{causal} = 0.2$. As shown in Figs 1E and 1F, VC-TWAS using summary-level GWAS data (VC-TWAS-SS) performed equivalent as using individual-level data under both simulation models. Additionally, VC-TWAS using filtered DPR weights and summary-level GWAS data (VC-TWAS_f) still had similar performance to that using complete DPR weights.

Fifth, we also compared VC-TWAS with CoMM [20,21] using individual-level and CoMM-SS using summary-level GWAS data under the scenarios with $p_{causal} = 0.2$ (S2 Fig). Although CoMM and CoMM-SS outperformed VC-TWAS, CoMM costed up to 600x more computation time and CoMM-SS costed up to 250x more computation time than VC-TWAS for testing a gene with ~5K test SNPs (S1 Table). Take typical genes with ~2K-5K test SNPs as examples, VC-TWAS costed ~20s versus ~38,037s by CoMM using individual-level GWAS data, and VC-TWAS costed ~3s versus ~373s by CoMM-SS using summary-level GWAS data, by a single-thread computation with 4 cores (32GB memory) on a 2.10GHZ CPU (16-node Intel Xeon node). As the number of the SNPs in a gene increases, the difference in computation run times between VC-TWAS and CoMM/CoMM-SS become even more pronounced.

Last but not least, to evaluate type I error of Burden-TWAS and VC-TWAS, we conducted $10^6$ times simulations under the null hypothesis where phenotypes were not associated with genetic data of the test gene. Without loss of generality, we used gene expression data simulated with $p_{causal} = 0.2$, $h_e^2 = 0.1$ and generated phenotypes randomly from a $N(0,1)$ distribution. We evaluated type I errors (Table 1) with multiple significant levels ($10^{-2}$, $10^{-4}$, $2.5 \times 10^{-6}$), demonstrating that both TWAS approaches had type I errors well controlled across all considered significance levels. We also presented the quantile-quantile (QQ) plots of p-values by all methods in S4 Fig.

**Table 1. Type I errors under null simulation studies for Burden-TWAS and VC-TWAS with $p_{causal} = 0.2$, $h_e^2 = 0.1$, using variant weights given by DPR, filtered DPR, and PrediXcan.**

| Significance Level | Burden-TWAS | | | VC-TWAS | | |
|---|---|---|---|---|---|---|
| | DPR | Filtered DPR | PrediXcan | DPR | Filtered DPR | PrediXcan |
| $1.00 \times 10^{-2}$ | $9.82 \times 10^{-3}$ | $9.86 \times 10^{-3}$ | $9.27 \times 10^{-3}$ | $9.43 \times 10^{-3}$ | $9.46 \times 10^{-3}$ | $9.23 \times 10^{-3}$ |
| $1.00 \times 10^{-4}$ | $8.64 \times 10^{-5}$ | $8.44 \times 10^{-5}$ | $9.95 \times 10^{-5}$ | $8.64 \times 10^{-5}$ | $9.05 \times 10^{-5}$ | $8.24 \times 10^{-5}$ |
| $2.50 \times 10^{-6}$ | $2.00 \times 10^{-6}$ | $2.00 \times 10^{-6}$ | $2.00 \times 10^{-6}$ | $2.00 \times 10^{-6}$ | $1.00 \times 10^{-6}$ | $6.00 \times 10^{-6}$ |

To summarize, VC-TWAS performed similarly to Burden-TWAS for studying phenotypes simulated from Model I with sparse true causal eQTL, but outperformed Burden-TWAS for studying phenotypes simulated from model II. This is because the genetic architecture assumed under Model I is the one assumed by Burden-TWAS with linear relationship between SNP effect sizes on phenotype and cis-eQTL effect sizes. Whereas, Model II assumes a genetic architecture that is a function of cis-eQTL effect sizes but not explicitly a linear relationship (leading to more general models of effect). Generally, TWAS methods using DPR weights achieved higher power than using PrediXcan weights when $p_{causal} \geq 0.01$, which is consistent with previous studies [8]. Additionally, VC-TWAS using filtered DPR weights achieved similar power as using complete DPR weights, while saving up to 80% of computation time.

## Application studies of AD with individual-level GWAS data

We applied VC-TWAS to the individual-level GWAS data of ROS/MAP and Mayo Clinic LOAD cohorts, using SNP weights (i.e., cis-eQTL effect sizes) generated by PrediXcan and filtered DPR methods with 499 ROS/MAP training samples that had both transcriptomic and genetic data profiled [8]. As suggested by previous studies [10,41], our TWAS results included genes with 5-fold cross validation (CV) $R^2 > 0.005$ for predicting quantitative gene expression traits by either PrediXcan or DPR. We obtained VC-TWAS p-values for 5,710 genes using PrediXcan weights and 12,650 genes using filtered DPR weights. Here, we roughly chose the threshold $10^{-4}$ to filter DPR weights in our VC-TWAS such that on average the variance component test considered ~50% SNPs from the test gene region. Specifically, the median number of SNPs considered by VC-TWAS per gene is 2,872 for using filtered DPR weights and 6,632 for using complete DPR weights (S3 Fig).

Leveraging the clinical and postmortem AD data available in ROS/MAP, we were able to apply VC-TWAS to four clinical and pathologic AD phenotypes (S1D Text). We examined final cognitive status diagnosis of 1,436 decedents (AD Dementia (N = 609) versus No Dementia (N = 827)), as well as three postmortem AD phenotypes including: continuous outcomes of $\beta$-amyloid load (N = 1,294), PHFtau tangle density (tangles, N = 1,303), and a global AD pathology (N = 1,329). In the VC-TWAS of all four AD phenotypes, we adjusted for covariates of age, smoking status, sex, study group (ROS or MAP), education, and the top three principal components of ancestry.

With Mayo Clinic cohort, we conducted VC-TWAS for AD clinical diagnosis with 844 cases diagnosed with LOAD and 1,255 controls showed no signal of dementia, while adjusting for covariates age, sex, and top three principal components of ancestry. Since only the phenotype of AD clinical diagnosis was profiled by both ROS/MAP and Mayo Clinic cohorts (under different diagnosis criteria) and different sets of covariates were adjusted in VC-TWAS, we conducted meta-analysis with VC-TWAS summary statistics for each study by using Fisher's method (meta VC-TWAS) [42] to increase power with a larger sample size.

By meta VC-TWAS, we detected 13 significant risk genes with FDR < 0.05 that were located within ~2MB region around the well-known AD risk gene *APOE* on chromosome 19 (Fig 2A; Table 2). Seven of those significant genes were known risk genes of AD by previous GWAS (*CLASRP*, *TOMM40*, *MARK4*, *CLPTM1*, *CEACAM19*, *RELB*) [43,44] and Burden-TWAS using DPR weights (*TRAPPC6A*) [8].

As clinical AD diagnosis can be associated with AD related pathologies, in further analyses we investigated whether the genes found to be associated with clinical AD diagnosis (Table 2) were also associated with pathologic AD phenotypes. We examined the VC-TWAS p-values of these significant genes with respect to AD pathology phenotypes ($\beta$-amyloid, tangles and global AD pathology) (S2 Table; Fig 2B; S5 and S6 Figs). Interestingly, 5 out of these 13 genes had at least one VC-TWAS p-value <0.0013 (Bonferroni correction with respect to 13 genes

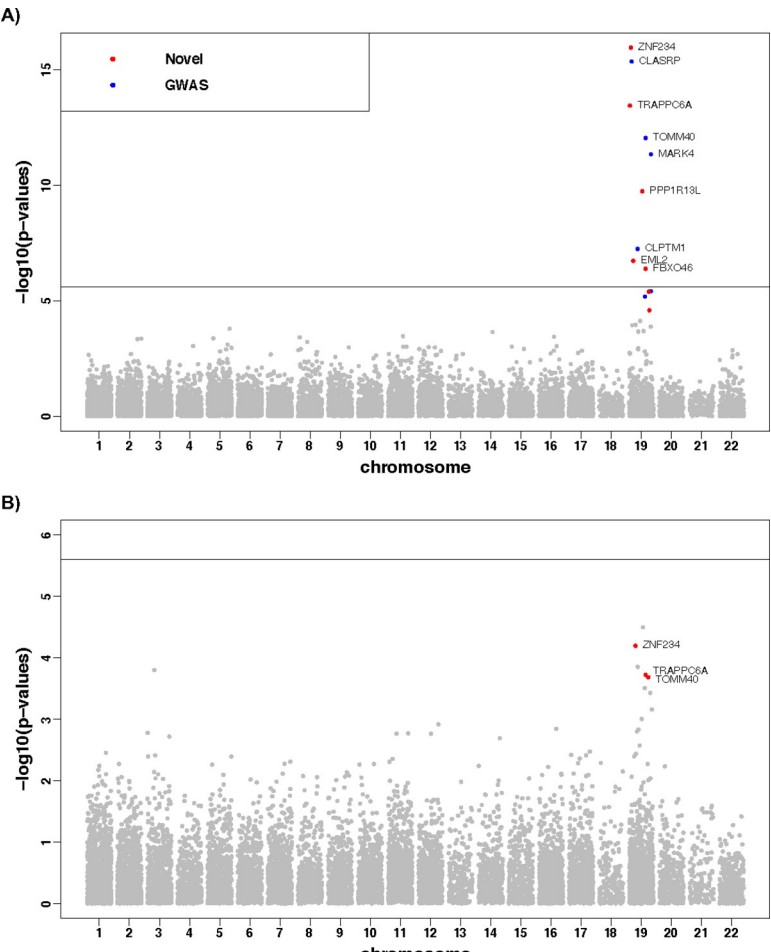

**Fig 2. Manhattan plots of meta VC-TWAS for AD clinical diagnosis (A) and VC-TWAS of global AD pathology (B) with filtered DPR weights.** Genes with FDR < 0.05 are colored in (A), with red for novel risk genes and blue for known AD risk genes. Genes with FDR < 0.05 in meta VC-TWAS of AD clinical diagnosis and p-value < 0.0013 in VC-TWAS of global AD pathology are colored in red in (B).

and 3 AD pathology phenotypes) for one of the three AD pathology phenotypes. Particularly, genes *ZNF234* and *CLASRP* have VC-TWAS p-value <0.0013 for all three AD pathology phenotypes, which are likely to be involved in the biological mechanisms of both β-amyloid and tangles. The other three genes (TRAPPC6A, *TOMM40*, *CEACAM19*) have VC-TWAS p-value <0.0013 for β-amyloid and global AD pathology, which are likely to be involved only in the biological mechanism of β-amyloid.

For example, the top significant gene *ZNF234* (with FDR = 1.40×10$^{-12}$) by meta VC-TWAS of AD clinical diagnosis is also the top significant gene (p-value = 2.10×10$^{-4}$) by VC-TWAS of β-amyloid, the second most significant gene (p-value = 6.39×10$^{-5}$) by VC-TWAS of global AD pathology, and has p-value = 1.06×10$^{-3}$ by VC-TWAS of tangles. These results showed that the genetic factor of gene *ZNF234* on AD could be potentially mediated through its gene expression, and the expression of this gene could be also potentially involved in the mechanisms of both AD pathology indices of β-amyloid and tangles. Besides AD, gene *ZNF234* is also a known risk gene for lipid traits [45]. The genetically regulated gene expression of this gene might also affect lipid traits, thus leading to a pleiotropy phenomenon of AD and lipid traits [46]. Additionally, *ZNF234* is known to be involved in the super pathway of gene expression

**Table 2. Significant genes for phenotype AD clinical diagnosis by meta VC-TWAS with filtered DPR weights.** Significant genes have FDR < 0.05 by meta-TWAS with ROS/MAP and Mayo Clinic cohorts. AD risk genes identified by previous GWAS are shaded in grey.

| Gene name | CHR | Start | End | P-value | FDR |
|-----------|-----|-------|-----|---------|-----|
| ZNF234[a] | 19 | 44,645,710 | 44,664,462 | $1.11 \times 10^{-16}$ | $1.40 \times 10^{-12}$ |
| CLASRP [a] | 19 | 45,542,298 | 45,574,214 | $4.44 \times 10^{-16}$ | $2.81 \times 10^{-12}$ |
| TRAPPC6A [a] | 19 | 45,666,187 | 45,681,485 | $3.60 \times 10^{-14}$ | $1.52 \times 10^{-10}$ |
| TOMM40 [a] | 19 | 45,394,477 | 45,406,935 | $9.05 \times 10^{-13}$ | $2.86 \times 10^{-9}$ |
| MARK4 | 19 | 45,754,550 | 45,808,541 | $4.62 \times 10^{-16}$ | $1.17 \times 10^{-8}$ |
| PPP1R13L | 19 | 45,882,892 | 45,909,607 | $1.82 \times 10^{-10}$ | $3.84 \times 10^{-7}$ |
| CLPTM1 | 19 | 45,457,848 | 45,496,598 | $5.71 \times 10^{-8}$ | $1.03 \times 10^{-4}$ |
| EML2 | 19 | 46,112,660 | 46,148,726 | $1.88 \times 10^{-7}$ | $2.97 \times 10^{-4}$ |
| FBXO46 | 19 | 46,213,887 | 46,234,151 | $4.13 \times 10^{-7}$ | $5.80 \times 10^{-4}$ |
| CEACAM19 [a] | 19 | 45,174,724 | 45,187,631 | $3.93 \times 10^{-6}$ | $4.68 \times 10^{-3}$ |
| GIPR | 19 | 46,171,502 | 46,185,704 | $4.07 \times 10^{-6}$ | $4.68 \times 10^{-3}$ |
| RELB | 19 | 45,504,695 | 45,541,452 | $6.63 \times 10^{-6}$ | $6.99 \times 10^{-3}$ |
| ZNF225 | 19 | 44,617,548 | 44,637,255 | $2.59 \times 10^{-5}$ | $2.51 \times 10^{-2}$ |

a: Genes with significant p-values <0.0013 (Bonferroni correction with respect to 13 genes and 3 phenotypes) for at least one AD pathology phenotype

and is annotated with the Gene Ontology term of nucleic acid binding and DNA-binding transcription factor activity [47].

Another significant gene of interest is *TOMM40*, which has FDR = $2.86 \times 10^{-9}$ by meta VC-TWAS for AD dementia and VC-TWAS p-values = ($4.44 \times 10^{-4}$, $6.95 \times 10^{-2}$, $1.91 \times 10^{-4}$) for $\beta$-amyloid, tangles, and global AD pathology, respectively. These findings suggest that the association of this well-known AD risk gene *TOMM40* [48] could be mediated through its gene expression via $\beta$-amyloid but not tangles.

For all SNPs considered by meta VC-TWAS of genes *ZNF234* and *TOMM40* for studying AD clinical diagnosis, we colocalized the meta GWAS results for AD clinical diagnosis with ROS/MAP and Mayo Clinic cohorts and the corresponding DPR weight (i.e., cis-eQTL effect size) magnitude. Interestingly, we found that the VC-TWAS association of these two genes were likely to be driven by SNPs around *APOE/TOMM40* loci that also possessed major cis-eQTL effect size magnitudes (S7 Fig).

In addition, our VC-TWAS identified a significant gene *HSPBAP1* (FDR = 0.058) for tangles (S5B Fig). As shown by previous studies, mRNA of gene *HSPBAP1* was abnormally expressed in the anterior temporal neocortex of patients with intractable epilepsy [49]. Based on our VC-TWAS results, gene *HSPBAP1* might not have a significant genetic effect on AD dementia, but might have a significant effect on tangle pathology (p-value = $4.57 \times 10^{-6}$) and may account for the increasing recognition of non-cognitive AD phenotypes [50]. This provides support about that gene *HSPBAP1* could be involved in the mechanism of brain pathology tangles and other neurological diseases such as intractable epilepsy [49,51].

We also applied VC-TWAS with PrediXcan weights, but no significant genes were identified with FDR <0.05 (S8, S9 and S10 Figs), neither the genes with smallest p-values were proximal to *APOE*. In contrast, our results by VC-TWAS with filtered DPR weights provided potential biological interpretations for several known AD risk genes via gene expression and for their associations with both clinical and pathologic AD phenotype.

## Application studies of AD with summary-level GWAS data

Next, we applied VC-TWAS to the stage1 summary-level GWAS data of AD from IGAP [3], which has a much larger sample size (~54K with 17,008 AD cases and 37,154 controls). We

considered filtered cis-eQTL DPR weights as used in the above applications with individual-level GWAS data, as well as the SNP weights of both cis- and trans- eQTL generated by the BGW-TWAS method [30]. LD covariance matrices from ROS/MAP individual-level GWAS data were used for implementing VC-TWAS with summary-level GWAS data.

**Using filtered cis-eQTL DPR weights.** By using the filtered cis-eQTL DPR weights, we identified 57 significant risk genes with FDR <0.05 by VC-TWAS (Fig 3A; S3 Table and S11A Fig), including the most well-known AD risk genes *TOMM40* and *APOE*, along with 45 genes

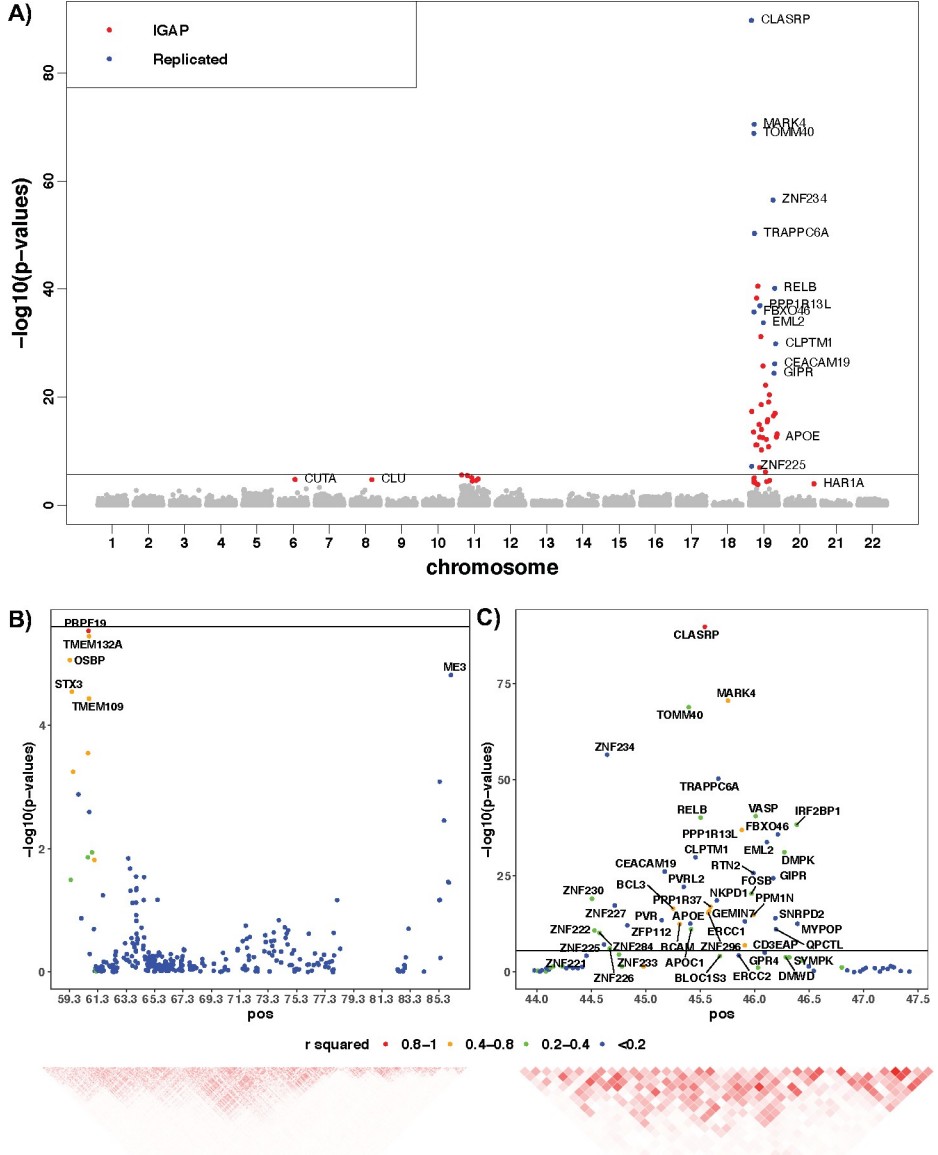

**Fig 3. Manhattan plots of VC-TWAS using IGAP summary data with filtered cis-eQTL DPR weights (A) and TWAS locus zoom plots for the loci on chromosome 11 (B) and chromosome 19 (C).** Significant genes with FDR < 0.05 are colored, with red for significant ones only identified by using IGAP summary data and blue for the ones replicating our VC-TWAS findings using the individual-level GWAS data of ROS/MAP and Mayo Clinic cohort. Between gene $R^2$ in (B, C) were calculated with respect to GReX values. The $R^2$ in locus zoom plot denoted by various colors for the dots is the squared correlation of GReX between the most significant gene and other neighborhood genes. The heatmap is based on the squared correlation matrix of GReX.

that were located within ~2MB region around those two risk genes. Other significant genes are distributed on chromosome 6, 8, 11 and 20 (Fig 3A). Among these 57 significant genes, 20 genes were known risk genes by previous GWAS [43,44] (S3 Table). Genes detected by VC-TWAS using the individual-level GWAS data of ROS/MAP and Mayo Clinic cohorts were all replicated (S3 Table). Burden-TWAS using IGAP summary data and filtered cis-eQTL DPR weights (S3 Table) also identified 22 out of 57 of these significant genes.

Since most significant genes are nearby genes on chromosome 11 and 19, we made analogous TWAS locus zoom plots for these two major loci in Fig 3B and 3C, where the between gene $R^2$ were calculated with respect to the predicted GReX values using the individual-level GWAS data of ROS/MAP and Mayo Clinic cohorts. We observed that most significant genes do have highly correlated GReX values, showing that nearby significant TWAS associations are likely to be not independent. For the locus on chromosome 19 (Fig 3C), we can see that the $R^2$ between gene *MARK4* (and a few other genes shown in orange color) and the top significant gene *CLASRP* is greater than 0.4. The significant genes in blue colors are likely to be independent associations from the top significant gene *CLASRP*. For the locus on chromosome 11 (Fig 3B), significant genes *TMEM132A*, *OSBP*, *STX3*, *and TMEM109* are highly correlated with the top significant gene *PRPF19*, while gene *ME3* tend to be another independent association.

**Using cis- and trans- eQTL weights generated by BGW-TWAS.** To provide a complementary list of significant genes by considering both cis- and trans- eQTL, we conducted VC-TWAS with the IGAP summary data using the cis- and trans- eQTL weights generated by the BGW-TWAS method [30]. We detected total 71 significant genes with FDR < 0.05 (Fig 4; S4 Table; S11B Fig), among which 6 genes were identified by GWAS [44,52–55] and 22 genes were shown to be related with AD or other neurological diseases by previous studies and (Table 3).

For example, gene *ARHGEF2* on chromosome 1(with FDR = 2.46×10$^{-13}$) is shown to be interact with all four members of the *MARK* family including *MARK4* on chromosome 19 which has an emerging tole in the phosphorylation of MAPT/TAU in Alzheimer's disease and identified by VC-TWAS using cis-eQTL DPR weights [56]. Gene *GAS5* (with FDR = 3.11×10$^{-19}$) was shown to have a novel role in microglial polarization and the pathogenesis of demyelinating diseases which suggested the potential therapeutic benefit of targeting *GAS5* for the

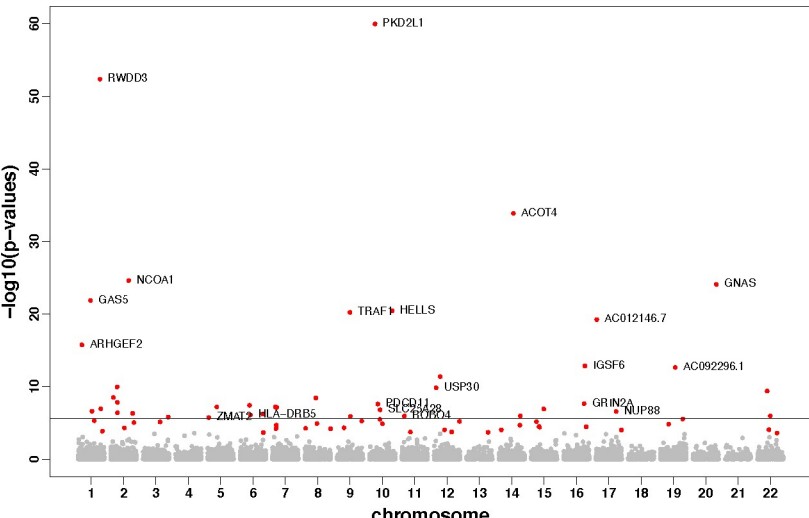

**Fig 4. Manhattan plot of VC-TWAS using IGAP summary data with BGW cis- and trans- eQTL weights.** Significant genes with FDR < 0.05 are colored.

**Table 3. Significant genes identified by VC-TWAS using IGAP summary statistics data with BGW cis- and trans- eQTL weights, which are either known GWAS loci or shown to be related with AD or other neurological diseases by previous studies.** AD risk genes identified by previous GWAS are shaded in grey.

| Gene Name | CHROM | Start | End | P-value | FDR |
|---|---|---|---|---|---|
| ARHGEF2[a] | 1 | 155,916,644 | 155,966,129 | $1.75 \times 10^{-16}$ | $2.46 \times 10^{-13}$ |
| GAS5[a] | 1 | 173,833,037 | 173,838,020 | $1.33 \times 10^{-22}$ | $3.11 \times 10^{-19}$ |
| NCOA1[a] | 2 | 24,714,782 | 24,993,571 | $2.42 \times 10^{-25}$ | $8.51 \times 10^{-22}$ |
| CENPO[a] | 2 | 25,016,004 | 25,045,245 | $4.07 \times 10^{-7}$ | $1.85 \times 10^{-4}$ |
| SLC4A10[a] | 2 | 162,280,842 | 162,841,792 | $8.80 \times 10^{-6}$ | $2.58 \times 10^{-3}$ |
| MAN2A1 | 5 | 109,025,066 | 109,205,326 | $6.30 \times 10^{-8}$ | $3.70 \times 10^{-5}$ |
| ZMAT2[a] | 5 | 140,079,746 | 140,086,266 | $1.83 \times 10^{-6}$ | $6.43 \times 10^{-4}$ |
| HLA-DRB5[a] | 6 | 32,485,119 | 32,498,064 | $7.81 \times 10^{-7}$ | $3.23 \times 10^{-4}$ |
| AHR[a] | 7 | 17,338,245 | 17,385,776 | $2.11 \times 10^{-5}$ | $5.62 \times 10^{-3}$ |
| JAZF1 | 7 | 27,870,191 | 28,220,362 | $7.49 \times 10^{-8}$ | $4.22 \times 10^{-5}$ |
| ACHE[a] | 7 | 100,487,614 | 100,494,594 | $5.93 \times 10^{-5}$ | $1.39 \times 10^{-2}$ |
| DLGAP2[a] | 8 | 1,449,530 | 1,656,642 | $1.18 \times 10^{-5}$ | $3.38 \times 10^{-3}$ |
| RFX3[a] | 9 | 3,218,296 | 3,526,004 | $4.69 \times 10^{-5}$ | $1.16 \times 10^{-2}$ |
| SLC25A28[a] | 10 | 101,370,281 | 101,380,535 | $1.55 \times 10^{-7}$ | $7.79 \times 10^{-5}$ |
| PDCD11[a] | 10 | 105,156,404 | 105,206,049 | $2.52 \times 10^{-8}$ | $1.69 \times 10^{-5}$ |
| ROBO4[a] | 11 | 124,753,586 | 124,768,396 | $1.13 \times 10^{-6}$ | $4.30 \times 10^{-4}$ |
| USP30[a] | 12 | 109,460,893 | 109,525,831 | $1.50 \times 10^{-10}$ | $1.41 \times 10^{-7}$ |
| PPP1R3E[a] | 14 | 23,765,111 | 23,772,057 | $2.11 \times 10^{-5}$ | $5.62 \times 10^{-3}$ |
| MTFMT[a] | 15 | 65,294,844 | 65,321,977 | $3.10 \times 10^{-5}$ | $8.09 \times 10^{-3}$ |
| PARP6 | 15 | 72,533,521 | 72,565,340 | $6.77 \times 10^{-6}$ | $2.07 \times 10^{-3}$ |
| GRIN2A[a] | 16 | 9,852,375 | 10,276,611 | $2.22 \times 10^{-8}$ | $1.56 \times 10^{-5}$ |
| SETD6[a] | 16 | 58,549,382 | 58,554,431 | $3.44 \times 10^{-5}$ | $8.81 \times 10^{-3}$ |
| NUP88[a] | 17 | 5,264,257 | 5,323,480 | $2.65 \times 10^{-7}$ | $1.24 \times 10^{-4}$ |
| NOS2[a] | 17 | 26,083,791 | 26,127,555 | $9.55 \times 10^{-5}$ | $2.07 \times 10^{-2}$ |
| CEACAM19 | 19 | 45,174,723 | 45,187,631 | $1.46 \times 10^{-5}$ | $4.03 \times 10^{-3}$ |
| APOC1[a] | 19 | 45,417,920 | 45,422,606 | $2.80 \times 10^{-6}$ | $9.63 \times 10^{-4}$ |

a: Genes shown to be related with AD or other neurological diseases by previous studies

treatment of neurological disorders including AD [57]. Gene *NCOA1* (with FDR $8.51 \times 10^{-22}$) was shown to have pivotal roles in memory and learning [58]. SNPs in Gene *PDCD11* (with FDR = $1.69 \times 10^{-5}$) have been showed to be significantly associated with AD [59]. Gene *APOC1* was detected by several GWASs [43,44], even when conditioning on *APOE* gene [60]. The presence of gene *ACHE* (with FDR = $1.39 \times 10^{-2}$) is a common feature in AD brain [61]. We also detected genes *NUP88* [62], *ROBO4* [63], *DLGAP2* [64], *AHR* [65], *PPP1R3E* [66], *RFX3* [67] and *NOS2* [68] that were shown to possess biological link with AD. Additional identified genes *USP30* [69], *GRIN2A* [70], *SLC25A28* [71], *HLA-DRB5* [72], *ZMAT2* [73], *SLC4A10* [74], *MTFMT* [75] and *SETD6* [76] were shown to be related with neurological diseases by previous studies.

Overall, we show that VC-TWAS using BGW weights provide a complementary list of risk genes to those previously identified by considering additional trans- eQTL information.

## Discussion

In this paper, we propose a novel variance-component TWAS (VC-TWAS) method that leverages eQTL effect sizes from reference transcriptome but does not assume a linear relationship between SNP effect sizes on phenotypes and cis-eQTL effect sizes. VC-TWAS is applicable to

both quantitative and dichotomous outcomes and can further handle both individual-level and summary-level GWAS data. By implementing this VC-TWAS with cis-eQTL effect sizes estimated by DPR method [8,13], we created a powerful test statistic that had good performance in simulation studies and obtained biologically meaningful TWAS results for both clinical and pathologic AD phenotypes. In particular, with the individual-level GWAS data of ~3.4K samples, we detected 13 TWAS genes for AD dementia, including the well-known GWAS risk gene *TOMM40* and previously identified TWAS gene *TRAPPC6A* [8]. Moreover, 6 out of these 13 genes were identified by previous GWAS [43]. The pleiotropy effects of 5 of these genes with respect to AD dementia and indices of AD pathology demonstrated the possible biological mechanisms linking AD risk genes via *β*-amyloid and tangles with AD dementia.

By applying VC-TWAS with summary-level GWAS data of AD with a much larger sample size, we not only validated our findings with the individual-level GWAS data but also detected additional novel risk genes. In particular, by applying VC-TWAS with both cis- and trans-eQTL effect sizes estimated by the BGW-TWAS method, we identified a list of significant AD risk genes that complement risk genes identified by considering only cis-eQTL information.

To help users conduct our VC-TWAS method conveniently and efficiently, we added this method into our previously developed tool––Transcriptome Integrated Genetic Association Resource (TIGAR) [8]. The user has the option of estimating cis-eQTL effect sizes by either the PrediXcan method (i.e., Elastic-Net) [12] or nonparametric Bayesian DPR method [13], or using eQTL weights generated by previous studies. The VC-TWAS function works for using individual-level GWAS data to study continuous or dichotomous phenotypes, as well as using summary-level GWAS data.

Since the variance component test statistic used by VC-TWAS involves calculating and performing an eigen-decomposition of a genotypic kernel matrix, efficient computation is required (even when filtering variants to include only those variants with relatively larger magnitude of eQTL estimates) for obtaining the corresponding p-values in practice. Our TIGAR tool implements multi-threaded computation to take advantage of high-performance cloud computing clusters and enable practical computation for testing genome-wide genes with respect to reference transcriptome data of multiple tissue types. A typical VC-TWAS with ~20K genes can be accomplished within ~80 hours by using a single thread with 1 CPU core and 32 GB memory.

Of course, current TWAS methods including our VC-TWAS still have their limitations. Because of genetic and transcriptomic heterogeneities across different ethnicities, one may have difficulty in translating eQTL effect size estimates across cohorts with different ethnicities [77]. That being said, VC-TWAS is likely more robust to this phenomenon than Burden-TWAS given the former relaxes the assumption of the latter of a linear relationship between the SNP effect size and eQTL effect in the transcriptome. Nevertheless, reference panels with diverse ethnicities and multiple tissue types are certainly needed to enhance TWAS to study complex diseases across different ancestral groups. We should note though that both ROS/MAP and Mayo Clinic cohorts that we considered in this work have similar (European) origins.

Compared to the alternative CoMM and PMR-Egger methods that jointly model both reference and test data, VC-TWAS might be less powerful when the eQTL effect sizes are homogeneous in both reference and test cohorts. However, CoMM and PMR-Egger methods are derived for quantitative phenotypes and implement likelihood ratio tests under a maximum likelihood reference framework that are computationally expensive when thousands of cis-SNPs need to be tested per gene. Although one may apply CoMM and PMR-Egger methods to dichotomous phenotypes by taking cases as 1's and controls as 0's, this may lead to inflated type I errors when population stratification leads to violation of the constant-residual variance

assumption. The recently published generalized linear mixed model association test (GMMAT) method paper has shown this by both simulation and real studies and suggested that a logistic mixed model would be more appropriate for analyzing dichotomous traits [78]. Thus, VC-TWAS would be preferred because of computational efficiency when imputed or whole genome sequencing genotype data are considered for the reference panel, or when dichotomous traits are analyzed. We provide a summary table for pros and cons of existing popular TWAS methods in S5 Table.

Nevertheless, our novel technique allows a more flexible framework to account for the unknown genetic architectures underlying the relationships between SNPs and the phenotype of interest with efficient computation. Using simulated and real AD-related data, we show our method VC-TWAS provides the public a useful tool for illustrating the genetic etiology of complex diseases by providing a list of risk genes whose effects on phenotypes might be mediated through transcriptomes.

## Web resources

VC-TWAS, https://github.com/yanglab-emory/VC_TWAS
TIGAR, https://github.com/yanglab-emory/TIGAR
PrediXcan, https://github.com/hakyim/PrediXcan
RADC Research Resource Sharing Hub, http://www.radc.rush.edu/
ROS/MAP data, https://www.synapse.org/#!Synapse:syn3219045
MayoClinicLOAD data, https://www.synapse.org/#!Synapse:syn2910256
GWAS catalog, https://www.ebi.ac.uk/gwas/
BGW weights of brain frontal cortex tissue, https://www.synapse.org/#!Synapse:syn22316792

## Supporting information

**S1 Text.** Details about PrediXcan's and TIGAR's approach of estimating cis-eQTL effect sizes (A, B), VC-TWAS approach with summary-level GWAS data (C), and ROS/MAP data (D). (DOCX)

**S1 Fig.** TWAS power comparison for VC-TWAS and Burden-TWAS with phenotypes simulated from Model I (A) and Model II (B). Various types of SNP weights were considered, including those derived from PrediXcan method, DPR method, and filtered DPR weights. In Model I, the combinations of causal probability and phenotype heritability are $(p_{causal}, h_p^2) = ((0.001, 0.2), (0.01, 0.3), (0.1, 0.4), (0.2, 0.5))$. In Model II, the combinations of causal probability and phenotype heritability are $(p_{causal}, h_p^2) = ((0.001, 0.1), (0.01, 0.1), (0.1, 0.15), (0.2, 0.15))$. (PDF)

**S2 Fig.** TWAS power comparison for VC-TWAS and CoMM with phenotypes simulated from Model I (A) and Model II (B) using individual-level and summary-level data under the scenarios with $p_{causal} = 0.2$. (PDF)

**S3 Fig. Box plot of the number of test SNPs considered by VC-TWAS of all genome-wide genes in the application studies of AD, with complete DPR weights and filtered DPR weights derived from the ROS/MAP training data.** (PDF)

**S4 Fig. Q-Q plots for VC-TWAS and Burden-TWAS with DPR weights, filtered DPR weights, and PrediXcan weights under null hypothesis, where quantitative gene expression traits were generated with $p_{causal}$ = 0.2 and $h_e^2$ = 0.1.**
(PDF)

**S5 Fig.** Manhattan plots of VC-TWAS results with filtered DPR weights for studying quantitative AD pathology of $\beta$-Amyloid (A) and tangles (B). Genes with FDA < 0.05 by meta VC-TWAS for studying AD clinical diagnosis are colored in red in (A) and top significant gene for studying tangles phenotype with FDR = 0.058 is colored in red in (B).
(PDF)

**S6 Fig. Q-Q plots of VC-TWAS results with filtered DPR weights for studying $\beta$-amyloid, tangles, and global AD pathology with ROS/MAP cohort, as well as meta VC-TWAS results with filtered DPR weights for studying AD clinical diagnosis with ROS/MAP and Mayo Clinic cohorts.**
(PDF)

**S7 Fig.** Locus zoom plots of GWAS results and the magnitude (i.e., absolute value) of cis-eQTL effect size estimates by DPR method for SNPs that were considered by VC-TWAS of genes *TOMM40* (A, B) and *ZNF2334* (C, D). Filtered test SNPs with the cis-eQTL effect size magnitude > $10^{-4}$ were plotted here. SNPs with GWAS p-value < $5 \times 10^{-8}$ were colored in red in (B,D), top significant SNPs by GWAS in (A,C) were shown as the blue triangle in (B,D).
(PDF)

**S8 Fig.** Manhattan plots of VC-TWAS results with PrediXcan weights for studying AD clinical diagnosis (A) and global AD pathology (B).
(PDF)

**S9 Fig.** Manhattan plots of VC-TWAS results with PrediXcan weights for studying quantitative AD pathology of β-Amyloid (A) and tangles (B).
(PDF)

**S10 Fig. Q-Q plots of VC-TWAS results with PrediXcan weights for studying $\beta$-amyloid, tangles, and global AD pathology with ROS/MAO cohort, as well as meta VC-TWAS results with PrediXcan weights for studying AD clinical diagnosis with ROS/MAP and Mayo Clinic cohorts.**
(PDF)

**S11 Fig. Q-Q plots of VC-TWAS results with cis-eQTL DPR filtered weights and BGW weights on IGAP GWAS summary statistics.**
(PDF)

**S1 Table. Average computation time per gene (in the unit of second) by all methods with individual and summary-level GWAS data by a single thread with 4 cores (32GB memory) from a 2.10GHZ CPU 16-core Intel Xeon computation node, for example genes that have test SNP numbers in respective range.**
(DOCX)

**S2 Table. Genes with VC-TWAS p-value <0.0013 with respect to at least one AD pathology phenotype and FDR <0.05 by meta VC-TWAS of AD clinical diagnosis.** AD risk genes identified by previous GWAS are shaded in grey.
(DOCX)

**S3 Table. Significant genes identified by VC-TWAS using IGAP summary statistics with filtered cis-eQTL DPR weights.** Significant genes were identified with FDR < 0.05. AD risk genes identified by previous GWAS are shaded in grey.
(DOCX)

**S4 Table. Novel significant genes identified by VC-TWAS using summary statistics with BGW weights on IGAP summary statistics.**
(DOCX)

**S5 Table. Pros and cons of existing popular TWAS methods.**
(DOCX)

## Acknowledgments

ROS/MAP study data were provided by the Rush Alzheimer's Disease Center, Rush University Medical Center, Chicago, IL. The MCADGC led by Dr. Nilüfer Ertekin-Taner and Dr. Steven G. Younkin, Mayo Clinic, Jacksonville, FL uses samples from the Mayo Clinic Study of Aging, the Mayo Clinic Alzheimer's Disease Research Center, and the Mayo Clinic Brain Bank.

## Author Contributions

**Conceptualization:** Michael P. Epstein, Jingjing Yang.

**Data curation:** Shizhen Tang, Aron S. Buchman, Philip L. De Jager, David A. Bennett.

**Formal analysis:** Shizhen Tang.

**Funding acquisition:** Jingjing Yang.

**Methodology:** Shizhen Tang, Michael P. Epstein, Jingjing Yang.

**Software:** Shizhen Tang.

**Writing – original draft:** Shizhen Tang.

**Writing – review & editing:** Aron S. Buchman, David A. Bennett, Michael P. Epstein, Jingjing Yang.

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
