## [Decision Letter · Decision Letter 0]

15 Sep 2020

Dear Dr Yang,

Thank you very much for submitting your Research Article entitled 'Novel Variance-Component TWAS method for studying complex human diseases with applications to Alzheimer's dementia' to PLOS Genetics. Your manuscript was fully evaluated at the editorial level and by independent peer reviewers. The reviewers appreciated the attention to an important problem, but raised some substantial concerns about the current manuscript. Based on the reviews, we will not be able to accept this version of the manuscript, but we would be willing to review again a much-revised version. We cannot, of course, promise publication at that time.

The manuscript proposed a variance-component-based TWAS method and applied it to study Alzheimer's disease.  Considering that there are many recent TWAS methods proposed in the literature, one theme of the reviewers is that additional comparisons with more recent methods are needed. In addition, one reviewer raised some important points regarding the application to ROSMAP and Mayo Clinic data which must be addressed in the revision.

If you decide to revise the manuscript for further consideration at PLOS Genetics, please aim to resubmit within the next 60 days, unless it will take extra time to address the concerns of the reviewers, in which case we would appreciate an expected resubmission date by email to plosgenetics@plos.org.

[LINK]

We are sorry that we cannot be more positive about your manuscript at this stage. Please do not hesitate to contact us if you have any concerns or questions.

Yours sincerely,

Lin Chen, Ph.D.

Guest Editor

PLOS Genetics

David Balding

Section Editor: Methods

PLOS Genetics

The manuscript proposed a variance-component-based TWAS method and have applied it to study Alzheimer's disease. Three reviewers have read the work and made some suggestions. Considering that there are many recent TWAS methods proposed in the literature, additional comparisons with more recent methods would be helpful. Moreover, one reviewer raised some valid points regarding the application to ROSMAP and Mayo Clinic data. This should be clarified and addressed in the revision.

Reviewer's Responses to Questions

**Comments to the Authors:**

Reviewer #1: Please see the attachment for comments

Reviewer #2: This paper proposes a novel Variance-Component TWAS procedure (VC-TWAS) that assumes the effects of cis-eQTL SNPs on phenotype are random (with variance proportional to corresponding cis-eQTL effects in reference dataset) rather than fixed. They further applied VC-TWAS using cis-eQTL effect sizes estimated by a nonparametric Bayesian method to study Alzheimer’s dementia (AD) related phenotypes and detected 13 genes significantly associated with AD, including 6 known GWAS risk loci such as TOMM40. It is an interesting problem that authors propose variance component tests to perform TWAS. Here are my major comments.

1. First, I think authors should compare other alternative methods in TWAS as there are several methods that may outperform standard TWAS, e.g., [1] and [2]. Authors need to numerically compare VC-TWAS these methods as well. Particularly, [1] connects CoMM with other SKAT and PrediXcan, showing how the improvement comes from (Section 2.4 in [1]). Since the test for equation (8) is closely related to equation (14) in [1], authors should cite the relevant reference for their works.

2. In the simulation studies, what are the sample sizes for equation (9)? As sample sizes for eqtl studies increase, the difference between PrediXcan and other refined ones becomes similar. In addition, Table 1 that summarizes the type I error is not enough especially for genome-wide scan. qq-plots for null simulations are more appropriate for type I error control.

3. In abstract, it is too strong to state that VC-TWAS gain > 2X power. This can be only done by simulation studies with some favorable settings. I am wondering why authors did not apply alternative methods to the real data analysis. The comparison in the real data analysis will be more interesting.

[1] Yang C, Wan X, Lin X, et al. CoMM: a collaborative mixed model to dissecting genetic contributions to complex traits by leveraging regulatory information[J]. Bioinformatics, 2019, 35(10): 1644-1652.

[2] Nagpal S, Meng X, Epstein M P, et al. TIGAR: an improved Bayesian tool for transcriptomic data imputation enhances gene mapping of complex traits[J]. The American Journal of Human Genetics, 2019, 105(2): 258-266.

Reviewer #3: Tang et al. proposed a TWAS method following the idea of the gene-based variance component test SKAT with cis-variant weights trained with Dirichlet process regression (DPR). The idea of combining TWAS and SKAT is novel, and the applications on Alzheimer’s disease data are interesting. The manuscript is also clearly written. Below I list some comments that may further improve the paper.

Major comments:

“As DPR method produces non-zero cis-eQTL effect size estimates for almost all SNPs within a test gene region (with most cis-eQTL effect size estimates being close to zero [8]), we explored an alternate VC-TWAS that considered a reduced set of SNPs by filtering out those with cis-eQTL effect size estimates smaller than the median cis-eQTL effect size estimate.”

1) Is the median cis-eQTL effect size estimate for each gene or all genes?

2) Other than using this arbitrary threshold for filtering, a method BGW-TWAS developed earlier by the authors’ group adopts a spike-and-slab prior that is commonly used in Bayesian variable selection. I am wondering what the performance of DPR as compared to the spike-and-slab prior.

3) Additionally, BGW-TWAS considers both cis and trans- eQTLs and has trained the same datasets as this manuscript. It would be interesting to use the BGW-TWAS trained cis- and trans- eQTL weights in this VC-TWAS approach. This will help compare: i) DPR vs. the spike-and-slab prior, and ii) cis- only vs. both cis- and trans- eQTLs.

4) Given that the main contribution of this manuscript is to use different SNP weights, I am curious about the real data results using the original SKAT to see the gain by using this new weighting via TWAS.

“Mayo Clinic LOAD GWAS data contain samples from two clinical AD Case-Control series:

Mayo Clinic Jacksonville (MCJ: 353 AD cases and 331 Controls), Mayo Clinic Rochester (MCR: 291 AD cases and 787 Controls) and a neuropathological series of autopsy-confirmed subjects from the Mayo Clinic Brain Bank (MCBB: 298 AD cases and 223 non-AD Controls) [24, 25]. In total, we have 844 cases with LOAD and 1,255 controls without a dementia diagnosis.”

1) This part was from the data description on the Synapse website, but the sum of the sample size of the three cohorts does not equal to the total number listed above. Specifically, the sample size of MCR and AD cases of MCBB does not match with those reported in Allen et al. Scientific Data 2016.

2) In the real data analysis of Mayo Clinic data, the authors probably need to adjust for the cohort variable. MCJ and MCR samples are antemortem, but MCBB samples are postmortem.

It is not clear if the outcome/AD definition is the same across ROSMAP data and the three cohorts of Mayo Clinic data. Below is an excerpt on the AD definition of Mayo data copied from the Synapse website.

“All subjects from the clinical series (MCJ and MCR) were diagnosed by a Mayo Clinic neurologist; all control subjects had a Clinical Dementia Rating score of zero at the most recent time of testing; all LOAD patients had a diagnosis of probable or possible AD according to the NINCDS-ADRDA criteria. Subjects selected from the Mayo Clinic Brain Bank underwent neuropathologic evaluation by Dr. Dennis Dickson. All ADs had definite diagnosis according to the NINCDS-ADRDA criteria and had Braak scores of ≥4.0. All non–AD Controls had Braak scores of ≤2.5; many had brain pathology unrelated to AD and, in particular, many of the non-AD samples used in the brain expression data were selected based on the presence of primary tauopathy (progressive supranuclear palsy).”

1) Were all AD patients in both ROSMAP and Mayo LOAD GWAS data diagnosed by a neurologist with the same criterion?

2) Even if 1) is true, the diagnosis of subjects in Mayo data has been assisted with NINCDS-ADRDA criteria and Braak score. For instance, “all LOAD patients had a diagnosis of probable or possible AD according to the NINCDS-ADRDA criteria.” Is this true for ROSMAP data? “All MCBB ADs had definite diagnosis according to the NINCDS-ADRDA criteria and had Braak scores of ≥4.0, all non–AD Controls had Braak scores of ≤2.5” is this true for all other cohorts?

I know it may be hard to check this, but making sure each dataset has the same AD definition and matched other clinical variables would facilitate the interpretation of the meta-analysis.

3) The non-AD controls in MCBB have other pathologies. Is this true for other cohorts? Would this cofound the comparison as well?

Before meta-analysis is done, are there any overlapping signals between the two data cohorts: ROSMAP and Mayo LOAD GWAS? This would help check the reproducibility of results.

“11 of these genes were shown to possess significant pleiotropic effects on brain pathology phenotypes.”

This significance is based on a p-value threshold of 0.05 (Table 3, Figure 2B) and has not been adjusted for multiple testing of 11 genes and three phenotypes/outcomes.

Minor comments:

Typo: Page 15, first line, CPU *munities*;

Page 19, line 457, “using individual-level DATA for”? “Data” was omitted.

**Have all data underlying the figures and results presented in the manuscript been provided?**

Reviewer #1: Yes

Reviewer #2: Yes

Reviewer #3: None

PLOS authors have the option to publish the peer review history of their article (what does this mean?). If published, this will include your full peer review and any attached files.

Reviewer #1: No

Reviewer #2: No

Reviewer #3: No

---

## [Decision Letter · Decision Letter 1]

11 Feb 2021

Dear Dr Yang,

Thank you very much for submitting a revised version of your Research Article entitled 'Novel Variance-Component TWAS method for studying complex human diseases with applications to Alzheimer's dementia' to PLOS Genetics.

Although two reviewers are now satisfied, Reviewer 2 continues to have concerns, primarily relating to the accuracy and fairness of descriptions of and comparisons with the CoMM software.  We ask you to further revise your manuscript to address these concerns, which can involve either performing new comparisons or a further explanation of the reasons for the discrepancies that the reviewer has highlighted.  We encourage you also to make software for your analyses available as the reviewer requests.

Should you decide to revise the manuscript for further consideration here, you should address all the points made by the reviewer either through appropriate revisions or rebuttal in a response letter. We will also require a description of the changes you have made in the manuscript.

We hope that you will revise the manuscript for further consideration at PLOS Genetics, please aim to resubmit within the next 60 days, unless it will take extra time to address the concerns of the reviewers, in which case we would appreciate an expected resubmission date by email to plosgenetics@plos.org.

[LINK]

We are sorry that we cannot be more positive about your manuscript at this stage. Please do not hesitate to contact us if you have any concerns or questions.

Yours sincerely,

Lin Chen, Ph.D.

Guest Editor

PLOS Genetics

David Balding

Section Editor: Methods

PLOS Genetics

Reviewer's Responses to Questions

**Comments to the Authors:**

Reviewer #1: Thanks for the revisions, my questions and concerns are well addressed.

Reviewer #2: Thank authors for detailed and comprehensive replies. I still have some comments.

1. In practice, many TWAS methods consider cis-SNPs to be either upstream or downstream 500kb with the range of a gene and the number of cis-SNPs ranges from tens of a few hundreds. In Table S1, the number of cis-SNPs considered ranges from 1000 to >5000. First, authors need to justify that increasing cis-SNP number can enhance power. Otherwise, I don’t see any points to consider such a large number of cis-SNPs to compare the computational efficiency. At least, authors need to show that using such a large of cis-SNPs in VC-TWAS can outperform CoMM/CoMM-S2 that uses a few hundred cis-SNPs. I guess authors are using imputed SNPs in the analysis. If so, authors need to justify that using imputed SNPs can improve power in TWAS over the genotyped SNPs. Second, to further account for horizontal pleiotropy, there are couple of transcriptome-wide Mendelian randomization methods proposed, e.g., PMR-Egger [1] that primary assumes a burden assumption. It will be also interesting to compare performance of VC-TWAS with PMR-Egger as well. In PMR-Egger, the authors compare the computational efficiency in Table 2 [1], but Table S1 is significantly different from that one. This point should be addressed seriously by the authors.

Overall, without justification for a large of cis-SNPs as Table S1, it is pretty aggressive to state as in the paragraph between lines 358 and 365 (as well between lines 109 and 111), which are absolutely misleading for readers as CoMM was designed to consider only a few hundreds genotyped cis-SNPs, especially such conclusions are based on the incorrect Table S1. Moreover, readers need to know whether they should use imputed cis-SNP or genotyped cis-SNP. If the power from genotyped cis-SNP is comparable to that of imputed cis-SNP, there is no reason to make comparisons in Table S1. I suggest the authors revise the parts I mentioned here.

2. According to CoMM, “The analysis for an individual trait in the NFBC1966 dataset can be done around 2 min and around 18 min for an individual disease in the GERA dataset on a Linux platform with 2.6GHz intel Xeon CPU E5-2690 with 30 720 KB cache and 96 GB RAM (only 7.8 GB RAM used) on 24 cores.” In CoMM, the authors assume that the number of cis-SNPs is less than sample size which makes sense as the median of the number of cis-SNPs (500kb on each side) in GWAS is around 200. For this, CoMM solves the problem using the X’X, which X is genotype and the computation is in order less than O(p^3). If the number of cis-SNPs is larger than the sample size, it is optimal to solve using XX’ and the computation is at most O(n^3). This is IMPORTANT based on your assumptions for the number cis-SNPs. In Table S1, we should expect at most 73*5^3 = 9125 seconds for SNP number > 5000 as I explained above. But the reported time is 10 times that of expected. Again, the results in Table S1 here is significantly different from Table 2 reported in PMR-Egger [1]. Thus, it must be somethings wrong here and authors must correct any mistakes for this table. As PMR-Egger is an extension considering a fixed intercept (burden test), authors may compare it as well. For a fair comparison, authors need to state clearly for the assumptions that all methods made. It will be helpful to have a table of summary for pros, cons, and any assumptions among different methods.

3. Though CoMM was developed using linear models, it can still be applied to binary outcome. The differences between using linear and logistic regression would not be that much as long as the case-control ratio is too extreme which is the case in this study. At least, it does not hurt to compare. Moreover, the sentence in introduction between lines 108 and 109 should be rewritten. It is more informative to state the assumption of VC-TWAS as the binary outcome and CoMM as the continuous outcome. It is not necessary that a linear model cannot be applied to the studies with binary outcomes. Please revise the corresponding statement in the manuscript.

4. To promote reproducibility, all codes for results (tables and figures) in this manuscript should be provided, e.g. in github.

5. Provided the parallel implementation, it is computationally feasible using CoMM for large sample size as GERA (PMR-Egger even used UKB data) so it would be interesting to compare CoMM/CoMM-S2 as well as PMR-Egger with VC-TWAS in real data analysis. Even without parallelization, it would at most take 18/60*24=7.2 hours to complete analysis for a trait in GERA that is much larger than the sample used in this manuscript.

[1] Yuan, Z., Zhu, H., Zeng, P., Yang, S., Sun, S., Yang, C., ... & Zhou, X. (2020). Testing and controlling for horizontal pleiotropy with probabilistic Mendelian randomization in transcriptome-wide association studies. Nature communications, 11(1), 1-14.

Reviewer #3: none

**Have all data underlying the figures and results presented in the manuscript been provided?**

Reviewer #1: Yes

Reviewer #2: **No: **To promote reproducibility, please provides all codes for Figures and Tables.

Reviewer #3: None

PLOS authors have the option to publish the peer review history of their article (what does this mean?). If published, this will include your full peer review and any attached files.

Reviewer #1: No

Reviewer #2: No

Reviewer #3: No

---

## [Decision Letter · Decision Letter 2]

11 Mar 2021

Dear Dr Yang,

Thank you very much for submitting your Research Article entitled 'Novel Variance-Component TWAS method for studying complex human diseases with applications to Alzheimer's dementia' to PLOS Genetics.

Reviewer 2 has only minor suggestions that we believe can be easily addressed.  In particular we think that the table of pros and cons of different methods would be useful. Please address the reviewer comments, we don't plan to send it back to reviewers again and so we hope to move quickly to acceptance once we receive your revision.

Please also:

1) Provide a list of your responses to the review comments and a description of the changes you have made in the manuscript.

We hope to receive your revised manuscript very soon.  If you anticipate any delay beyond 30 days we ask you to let us know the expected resubmission date by email to plosgenetics@plos.org.

Please notify the journal office if the reviewer attachment is missing. They will also be available for download from the link below. You can use this link to log into the system when you are ready to submit a revised version, having first consulted our Submission Checklist.

[LINK]

Yours sincerely,

Lin Chen, Ph.D.

Guest Editor

PLOS Genetics

David Balding

Section Editor: Methods

PLOS Genetics

Reviewer's Responses to Questions

**Comments to the Authors:**

Reviewer #2: Attached

**Have all data underlying the figures and results presented in the manuscript been provided?**

Reviewer #2: Yes

PLOS authors have the option to publish the peer review history of their article (what does this mean?). If published, this will include your full peer review and any attached files.

Reviewer #2: No

---

## [Editor Report · Decision Letter 3]

15 Mar 2021

Dear Dr Yang,

We are pleased to inform you that your manuscript entitled "Novel Variance-Component TWAS method for studying complex human diseases with applications to Alzheimer's dementia" has been editorially accepted for publication in PLOS Genetics. Congratulations!

Yours sincerely,

Lin Chen, Ph.D.

Guest Editor

PLOS Genetics

David Balding

Section Editor: Methods

PLOS Genetics

Comments from the reviewers (if applicable):

**Data Deposition**

http://datadryad.org/submit?journalID=pgenetics&manu=PGENETICS-D-20-01014R3

**Press Queries**

---

## [Editor Report · Acceptance letter]

29 Mar 2021

PGENETICS-D-20-01014R3 

Novel Variance-Component TWAS method for studying complex human diseases with applications to Alzheimer's dementia 

Dear Dr Yang, 

We are pleased to inform you that your manuscript entitled "Novel Variance-Component TWAS method for studying complex human diseases with applications to Alzheimer's dementia" has been formally accepted for publication in PLOS Genetics! Your manuscript is now with our production department and you will be notified of the publication date in due course.

With kind regards,

Alice Ellingham

PLOS Genetics

On behalf of:
